# A pocket-escaping design to prevent the common interference with near-infrared fluorescent probes in vivo

Panfei Xing[1], Yiming Niu[1], Ruoyu Mu[1], Zhenzhen Wang[1,2], Daping Xie[1], Huanling Li[2], Lei Dong [2] & Chunming Wang [1✉]

Near-infrared (NIR) fluorescent probes are among the most attractive chemical tools for biomedical imaging. However, their in vivo applications are hindered by albumin binding, generating unspecific fluorescence that masks the specific signal from the analyte. Here, combining experimental and docking methods, we elucidate that the reason for this problem is an acceptor (A) group-mediated capture of the dyes into hydrophobic pockets of albumin. This pocket-capturing phenomenon commonly applies to dyes designed under the twisted intramolecular charge-transfer (TICT) principle and, therefore, represents a generic but previously unidentified backdoor problem. Accordingly, we create a new A group that avoids being trapped into the albumin pockets (pocket-escaping) and thereby construct a NIR probe, BNLBN, which effectively prevents this backdoor problem with increased imaging accuracy for liver fibrosis in vivo. Overall, our study explains and overcomes a fundamental problem for the in vivo application of a broad class of bioimaging tools.

[1] State Key Laboratory of Quality Research in Chinese Medicine, Institute of Chinese Medical Sciences, University of Macau, Macau, SAR, China. [2] State Key Laboratory of Pharmaceutical Biotechnology, Nanjing University, 210093 Nanjing, China. ✉email: cmwang@umac.mo

Fluorescent molecules with near-infrared (NIR) emission are among the most attractive chemical tools for biomedical imaging[1]. Because of their advantages such as deep tissue penetration and low photo cytotoxicity[2], an increasing number of NIR probes have been developed for noninvasive, real-time detection of various biomolecules in vivo[3]. However, recent findings reveal that NIR fluorophores/probes—especially those designed with a turn-on responsiveness from the twisted intra-molecular charge transfer (TICT) structure—can unexpectedly bind to serum albumin, and this binding generates fluorescence[4,5]. Since albumin is one of the most abundant proteins in the living system[6], an NIR probe used in vivo has a high chance to bind it, producing unspecific fluorescence to mask the specific signal from the analyte[7]. Some excellent studies took the advantage of this binding to prepare albumin-bound imaging tools with prolonged circulation or increased biocompatibility[8–14]; nevertheless, for NIR molecular probes designed for analyte-specific detection, the albumin interference remains a critical and common issue hampering their in vivo imaging accuracy.

To prevent this issue, it is necessary first to find the reason that albumin triggers unwanted fluorescence, which lies in the design principle of the fluorophores. Most organic NIR fluorophores, including cyanine, Nile blue, and dicyanomethylene-4H-pyran (DCM), are developed based on an acceptor−π−donor (A−π−D) structure with the TICT mechanism[15–17]. Such a fluorophore typically contains a pair of D and A group, which perform at least two fundamental roles. First, the electron transfer from D to A is the basis for fluorescent emission. Second, the twist of the D/A pair, together with a reasonably extended π-conjugation, enables the emission wavelength to fall into a desirable NIR region[18,19]. Typical strategies to develop NIR probes from fluorophores focuses on the first role, by manipulating the electron transfer from D to A (Fig. 1a). For instance, a recognition moiety (R) is modified to the D end to block its electron supply ("off"), before a specific analyte takes off (or changes) R to reactivate the D-to-A electron supply and emit fluorescence ("on")[20–25]. The status of the A end, or the influence of its change on fluorescence, is less noted. However, recent studies indicate that albumin can enhance the fluorescence of the TICT-based fluorophores by affecting the second role of the D/A pair. It binds the dye, restricts its D/A twist, and immobilizes the dye's molecular conformation[26,27]. Because when D/A twists, it is the more flexible A group—instead of the rigid D group—that rotates, it is possible that A is the target for albumin to bind. Because most attention in probe design focuses on modifying D, the vulnerability of A to the interference by albumin (and other possible molecules) emerges as a common-existing backdoor problem that potentially affects all TICT-based NIR probes used for in vivo imaging (Fig. 1b, c). In practice, preventing albumin binding may enable the probe to avoid fast hepatobiliary clearance and maintain its activity in physiological condition.

Here, we demonstrate a simple and generic approach to overcome this problem, by designing an NIR fluorophore with a coplanar A group that can avoid binding to albumin. First, combining experimental and molecular docking methods, we elucidated that the mechanism underlying albumin−fluorophore binding is that the former's hydrophobic pockets trap the latter's A group. Then, based on this finding, we designed a type of A group that avoids being trapped into the hydrophobic pockets of albumin ("pocket-escaping") and accordingly constructed an NIR probe. This probe effectively prevented albumin interference with improved accuracy for in vivo imaging in a murine liver fibrosis model.

## Results

### Albumin mediates fluorescence of TICT NIR fluorophores

To validate that it is a common issue that albumin can bind to and activate TICT-based NIR dyes, we synthesized five compounds, namely DCM, NLB, HCy7, Cy5, and Cy7 (Fig. 1d), which represent the three common types of DCM (DCM), Nile blue (NLB), and cyanides (HCy7, Cy5, and Cy7) dyes[16,28–30]. They are all constructed based on TICT mechanism but have different A/D groups to achieve fluorescence emission in the NIR zone. First, photophysical examination showed that albumin—of human (HSA), bovine (BSA), and mouse (MSA) origin—as well as fetal bovine serum (FBS) significantly enhanced the fluorescent signal of all the five compounds in phosphate buffer saline (PBS, pH 7.4, 10 mM, room temperature; Fig. 1d and Supplementary Figs. 1–6). A positive correlation was observed between the albumin concentration (10–600 μM) and the fluorescence enhancement factor.

To explore the possible reason for albumin's interaction with the dyes, we tested three other representative macromolecules in parallel. Among them, glycerol is commonly used for increasing viscosity and thereby restricting the molecular twisting of the bound compounds[31,32]; and heparin and polyvinyl alcohol (PVA) have a similar molecular weight with albumin but, not a transport protein, represent natural polysaccharides and synthetic polymers, respectively. Interestingly, glycerol (10–50% in PBS) also enhanced the fluorescence of these dyes, though its extent of enhancement was much less than that of albumin. However, raising the concentration of albumin did not increase the viscosity of its PBS solution—unlike glycerol (Fig. 1e). Meanwhile, heparin and PVA caused little influence on the fluorescence emission of the dyes. Instead, as revealed by [1]H NMR titration of albumin in different concentrations to Cy5S (a sulfated Cy5 derivative with higher hydrophilicity) in deuterium oxide ($D_2O$), the resonance of the A group (marked with red line) of Cy5S was weakened upon the addition of albumin (0.1–0.8 equivalent; Fig. 1f and Supplementary Fig. 8), until the proton signal of the indole acceptor completely disappeared. In summary, the above findings suggest that albumin enhances the fluorescence of common TICT-based NIR dyes, by targeting the twistable A group on the dye molecule.

### Albumin interacts with the dyes by capturing the A groups

Based on the above findings, we hypothesized that the albumin interacts with the TICT NIR dyes through the binding between the former's hydrophobic pockets and the latter's variable A groups that can rotate. To validate this hypothesis, we transplanted the A groups of the dyes onto a nonsensitive fluorophore backbone, coumarin 6H (Fig. 2a). Accordingly, these coumarin derivatives were named as NC, BNC, BNCQ, BNCN, BNCY, and BNCE (Fig. 2b), and their photophysical characteristics were examined (Table 1). For BNC, BNCQ, and BNCN as examples, the maximal emission wavelength was 402, 520, and 590 nm, with a visible emission color of blue, green, and red (Fig. 2c) and an energy gap of 4.34, 3.91, and 3.05 eV (Fig. 2d), respectively. Further, the rotation of the malononitrile moiety on BNCN (with a dihedral angle of 90°) created an asymmetric π-conjugated backbone (BNCN-90°) that resulted in charge redistribution and further induced TICT. The DFT data indicated that TICT lowered the energy gap from 3.05 to 0.94 eV. These results showed that both (i) extending π-conjunction (BNCQ) and (ii) transplanting A group on a π-conjunction (BNCN) are effective means to lower the highest occupied molecular orbital (HOMO) and the lowest unoccupied molecular orbital (LUMO) of a molecule. And the second means is perhaps more effective to red-shift the emission wavelength of a fluorophore by forming a TICT structure. These six compounds were treated with albumin in different concentrations (10–600 μM). Notably, NC, BNCN, BNCY, and BNCE showed a concentration-dependent enhancement in

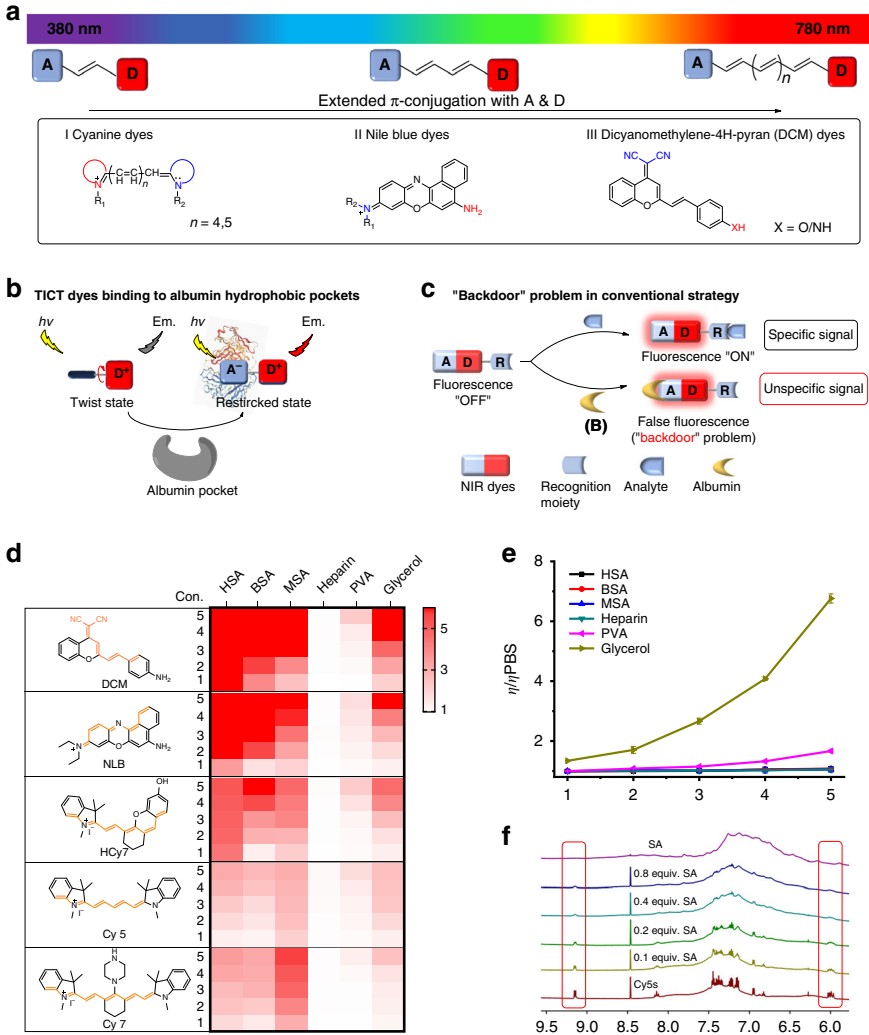

**Fig. 1 Validation of a common "backdoor" problem for NIR probes. a** Construction of typical NIR dyes based on the TICT mechanism, including (I) Cyanine dyes, (II) Nile blue dyes, and (III) DCM dyes, with their electron acceptor ("A") and donor ("D") groups illustrated. **b** Albumin can bind to, and restrict the twisting of, the A end of these TICT dyes, which **c** produces unspecific and interfering fluorescent signals that mask the specific and expected signals from the analyte. Because of the abundance of albumin in the living system, this issue represents a common, but largely overlooked, "backdoor" problem for the in vivo application of such probes. **d** The chemical structures of representative TICT-based NIR dyes and fold changes (from white to red) of these dyes in response to HSA, BSA, MSA, heparin, PVA, and glycerol in PBS (pH 7.4, 10 mM). The fluorescence enhancement for a dye is determined by the ratio of its fluorescent intensity in a different condition to that in PBS. Con. indicates the concentration of the analytes. **e** The relative viscosity of HSA (black), BSA (red), MSA (blue), heparin (dark cyan), PVA (magenta), and glycerol (dark yellow) in different concentrations in PBS (pH 7.4, 10 mM). Data are represented as mean values ± SD ($n = 3$ independent experiments). **f** $^1$H NMR titration of different equivalent albumin (0.1–0.8 equiv.) to Cy5S in $D_2O$ (from 6.0 to 9.5 ppm). Red frames highlight the proton signals on the A group. The concentrations of HSA, BSA, MSA, heparin, and PVA used: 1, 10 μM; 2, 50 μM; 3, 100 μM; 4, 300 μM; 5, 600 μM. The content of glycerol used: 1, 10%; 2, 20%; 3, 30%; 4, 40%; 5, 50%. Source data for Fig. 1d, e are provided as a Source Data file.

fluorescence spectrum with the addition of albumin (Fig. 2e and Supplementary Fig. 9), while no apparent changes were observed for BNC and BNCQ treated in the same way. Hence, implanting the different A groups switched the nonsensitive core (BNC) into a sensitive one, resulting from the interaction with albumin.

Construction of these molecules enabled us to perform a competition assay to reveal the interaction between albumin and the A groups. We chose five drugs known for site-specifically binding albumin: warfarin—site IIA; ibuprofen—site IIB and IIIA; indomethacin—IB and IIA; iodipamide—IIA; and propofol—site IIIA and IIIB[33]. We pretreated albumin solution with one of the above drugs (50–250 μM) to block the binding site, before the addition of NC, BNCN, BNCY, and BNCE. As shown in Fig. 2f and Supplementary Figs. 10–12, some of these drugs exhibited a concentration-dependent displacement of these TICT

dyes with albumin, determined by changes in the fluorescence intensity. These data indicated the binding sites on albumin for the dyes to be: (i) IB, IIA, and IIB for NC; (ii) IIA and IIB, for BNCN; (iii) IIA and IIB for BNCY; (iv) IB, IIA, and IIB for BNCE.

Outcomes from a docking assay further elucidated the mechanism. As shown in Supplementary Table 1, -CDOCKER energy (kcal/mol) indicates the energy of pocket−dye complexes, where a higher value reflects that the dye is harder to dissociate from the pocket. -CDOCKER interaction energy (kcal/mol) indicates the energy of the pocket−dye interaction, where a higher value reflects that the dye is easier to combine with the pocket. The values (~40 kcal/mol) suggest that these TICT-based dyes have the affinity to bind to the pockets of albumin, in agreement with the outcomes from the competition assay (Fig. 2f,

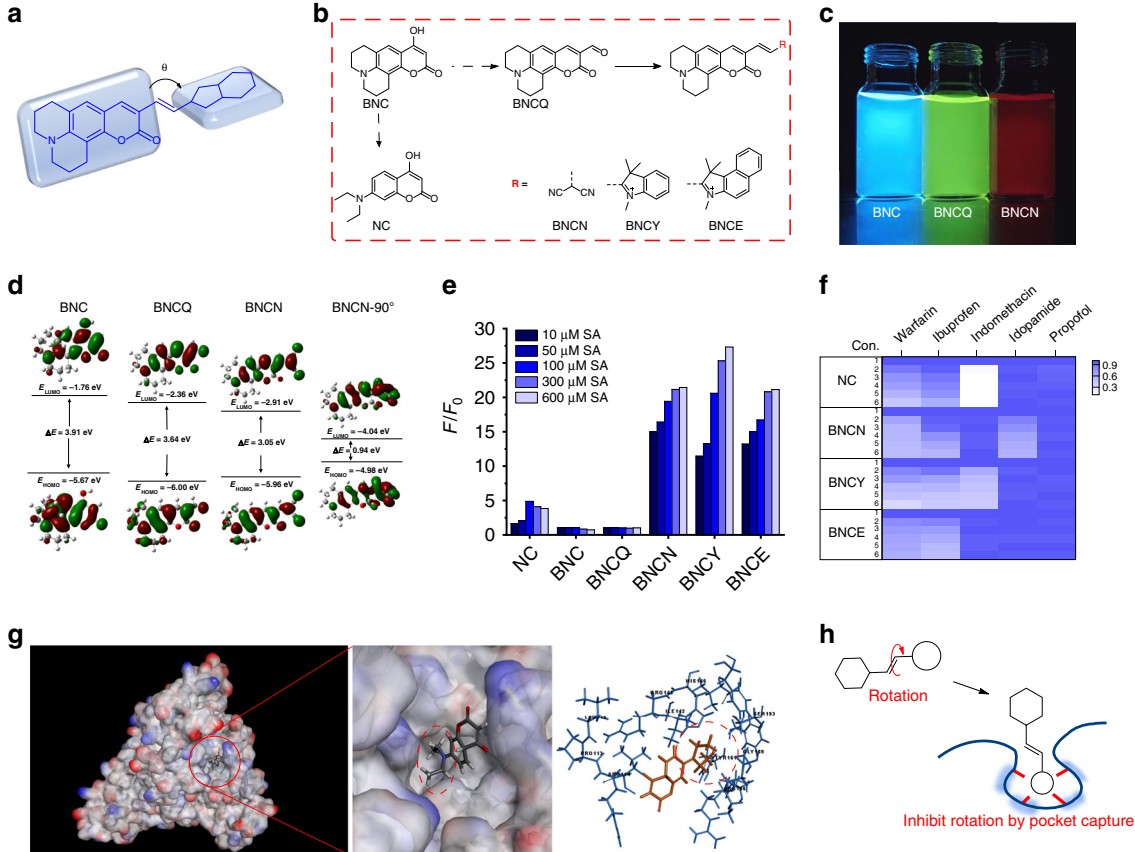

**Fig. 2 Elucidation of the albumin-fluorophore binding mechanism. a** The illustration of "transplanted" TICT dyes. **b** Chemical structures of NC, BNC, BNCQ, BNCN, BNCY, and BNCE. **c** Fluorescence of BNC, BNCQ, and BNCN in PBS solution (pH 7.4, 10 mM) under a 365 nm excitation. **d** DFT optimized molecular orbital plots (LUMO and HOMO) of BNC, BNCQ, BNCN, and BNCN-90°. **e** Concentration-dependent fluorescence changes of NC, BNC, BNCQ, BNCN, BNCY, and BNCE with the addition of albumin (10−600 μM) in PBS solution (pH 7.4, 10 mM) at room temperature. **f** Fold changes (from white to blue) for NC, BNCN, BNCY, and BNCE in 10 μM albumin with different concentration of inhibitors. Con. indicates the concentration of the inhibitors. 1, 0 μM; 2, 50 μM; 3, 100 μM; 4, 150 μM; 5, 200 μM; 6, 250 μM. **g** The conformation of albumin and NC. **h** The scheme of "pocket capture" for albumin to TICT dyes. $F_O$ and $F$ are the fluorescence intensity at the maximum emission wavelength without and with the presence of analyte, respectively. Source data for Fig. 2e, f are provided as a Source Data file.

**Table 1 Photophysical properties of NC, BNC, BNCQ, BNCN, BNCY, and BNCE in PBS solution at room temperature.**

| Dye | Absorbance $\lambda_{max}$ (nm) | Emission (nm) | Stoke shift (nm) | $\varepsilon$ (L M$^{-1}$ cm$^{-1}$) | Quantum yield (%) | $\Delta E$ (eV) |
|---|---|---|---|---|---|---|
| NC | 343 | 402 | 59 | 14,600 | 15.31 | 4.34 |
| BNC | 405 | 465 | 60 | 19,100 | 31.26 | 3.91 |
| BNCQ | 465 | 520 | 55 | 37,900 | 39.74 | 3.64 |
| BNCN | 555 | 590 | 35 | 26,500 | 3.69 | 3.05 |
| BNCY | 596 | 680 | 84 | 65,700 | 1.31 | 2.50 |
| BNCE | 599 | 690 | 91 | 35,200 | 1.22 | 2.39 |

The quantum yields were determined using cresol purple as reference ($\Phi_f = 0.58$ in ethanol).

Supplementary Figs. 10–12, and Supplementary Table 1). It is also found that BNCN and BNCE failed to bind site IIB and IB, respectively, mainly due to the steric and electronic factors. Notably, only the twistable A groups of these compounds were captured by the pockets of albumin, while the rigid π-conjunction parts harbored outside (Fig. 2g and Supplementary Figs. 10–12). Collectively, these findings suggest that the capture of the dye's A group into the albumin's hydrophobic pockets is the key mechanism for the binding (Fig. 2h). This capture restricts the innate rotation of the dye molecule, leading to a strong enhancement of fluorescence. This enhancement is less noticed,

because typical strategies for probe design focus on modifying the D end and manipulating the D−A electron transfer. Our findings uncover that affecting the twist of the A group ignites unspecific fluorescence, which can be a common backdoor problem for in vivo applications. The demonstration of the A-pocket interaction helped us devise an approach accordingly to prevent this problem.

**Designing pocket-escaping A group avoids albumin binding.** Inspired by the pocket capture mechanism between the A groups

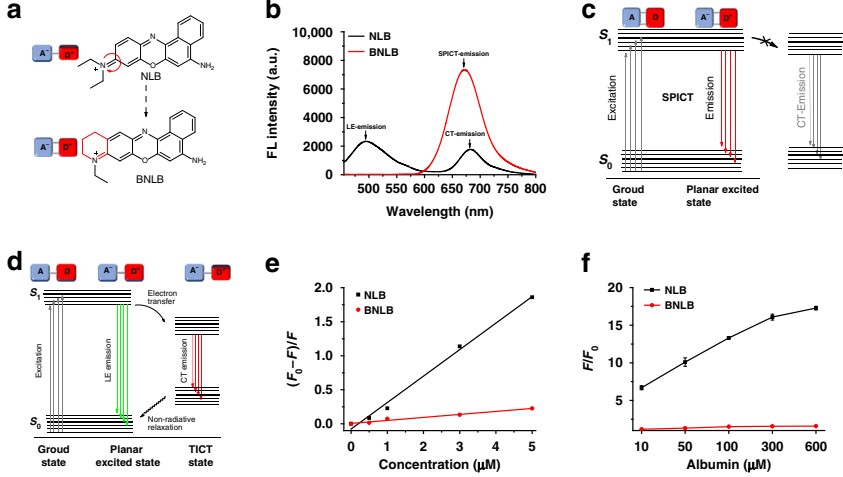

**Fig. 3 Design of "pocket-escaping" NIR dyes to avoid albumin binding. a** The chemical structure of BNLB. **b** Fluorescence emission of NLB (black) and BNLB (red) in PBS buffer solution (pH 7.4, 10 mM) with an excitation at 420 nm. **c**, **d** Jabłoński diagram showing the emission mechanism of BNLB and NLB. **e** Binding affinities of BNLB (red) and NLB (black) to albumin ($n = 3$ independent experiments). **f** Fluorescence enhancement of NLB (black) and BNLB (red) with the addition of albumin ($10-600 \, \mu$M). Data are represented as mean values ± SD ($n = 3$ independent experiments). $F_0$ and $F$ are the fluorescence intensity at the maximum emission wavelength without and with the presence of analyte, respectively. Source data for Fig. 3e, f are provided as a Source Data file.

of TICT dyes and the hydrophobic pockets of albumin, we conceived the idea of designing a pocket-escaping A group to avoid albumin binding and therefore interference. Based on NLB, we created BNLB as an improved NIR dye. The twistable A group ($N$, $N$-diethylamino moiety) on NLB was blocked in design, making BNLB a rigid coplanar π-conjugated structure (Fig. 3a). Upon excitation at 420 nm, NLB showed a characteristic TICT emission, including a local excited (LE) emission (490 nm) and a charge-transfer (CT) emission at NIR region (680 nm) (in PBS, pH 7.4, 10 mM). In comparison, BNLB showed a single NIR emission at 670 nm (Fig. 3b). We speculated that the emission was contributed by the coplanarity and rigidity of BNLB, which promote strong intermolecular electronic coupling and reduce the energy barrier for the intermolecular transport of charges—the two consequences further result in a stretched planar ICT (SPICT)[34]. This mechanism uniquely provides BNLB with the essential property of NIR emission, while avoiding the reliance on perpendicular twisting of the A moieties in this process—which is the reason for the unexpected fluorescence found in TICT dyes (Fig. 3c, d).

Our experiments confirmed that BNLB did not bind albumin as NLB did. First, the binding assay[35] indicated that BNLB had a much lower binding affinity for albumin ($K = 4400 \, \text{M}^{-1}$) compared with NLB ($K = 38,910 \, \text{M}^{-1}$, Fig. 3e). Second, as quantitatively measured by isothermal titration calorimetry (ITC), NLB bound to albumin with a calculated $K_d$ of 161 μM. In contrast, the interaction of BNLB with albumin was undetectable (Supplementary Fig. 13). Third, a native polyacrylamide gel electrophoresis (native PAGE) was especially employed to characterize the dye−protein binding (Supplementary Fig. 14), with strong fluorescence observed with the NLB−albumin complex but no signal detected in the BNLB−albumin sample. The above results are consistent with the fluorescence titration data of albumin to NLB and BNLB. Addition of albumin ($10-600 \, \mu$M) markedly enhanced the fluorescence of NLB but caused little change to that of BNLB (Fig. 3f). These data indicate that blocking the twistable A group on TICT dyes effectively facilitates the dye to escape the pocket capture by albumin. Consequently, the "pocket-escaping" BNLB should have the potential to avoid albumin interference and provide more accurate imaging effect in vivo, which we continued to validate below.

**Pocket-escaping increases accuracy for in vivo imaging**. To validate that pocket-escaping can effectively prevent unwanted fluorescence caused by albumin, we constructed an NIR probe with such feature and tested its efficacy in a murine liver fibrosis model. Allysine (α-aminoadipic-δ-semialdehyde), abundantly produced by lysyloxidase (LOX) during liver fibrosis, is a reliable pathological marker (Fig. 4a)[36–38]. To trace allysine, we introduced a hydrazine moiety onto the BNLB dye to develop a BNLBN probe. Hydrazine is highly nucleophilic to allysine and can quench the fluorescence of BNLB, through an acceptor-photoinduced electron transfer effect (a-PET) effect. When the a-PET effect is removed by the condensation reaction between hydrazine and allysine, the compound releases fluorescence (Fig. 4b).

In parallel, we synthesized NLBN to serve as the control (a TICT-based NIR probe) to BNLBN, by only modifying the recognition hydrazine moiety onto the D part of NLB. In both BNLBN and NLBN groups, addition of oxidized albumin (O-SA), which has allysine[39], triggered an obvious increase in fluorescence intensity within 6 min (Fig. 4c, d, Supplementary Figs. 16 and 17). However, in the NLBN group, addition of albumin also turned on fluorescence that was as strong as that caused by O-SA, suggesting that the use of NLBN for in vivo imaging would be heavily interfered by albumin. Meanwhile, albumin produced very low fluorescence with BNLBN, making the O-SA-triggered signal distinguishable.

We further constructed molecular logic gates to confirm that the albumin-generated signal is the backdoor for NLBN. Allysine (glutaraldehyde as a substitution) and albumin were selected as two inputs to the recognition moiety and the A part of NLBN, respectively. As shown in Fig. 4e, a superimposed effect in the output was observed upon the addition of two inputs (1,1), compared with the addition of allysine (1,0) or albumin (0,1), determined by its relative fluorescence intensity changes. Notably, the output for allysine (1,0) input was negligible, compared with that from the albumin (0,1) input, indicating this backdoor as a significant fluorescent switch.

Before starting the animal study, we performed cytotoxicity tests (CCK-8 assays) and confirmed that both BNLBN and NLBN were nontoxic (viability >90% at 100 μM after 24 h) to two

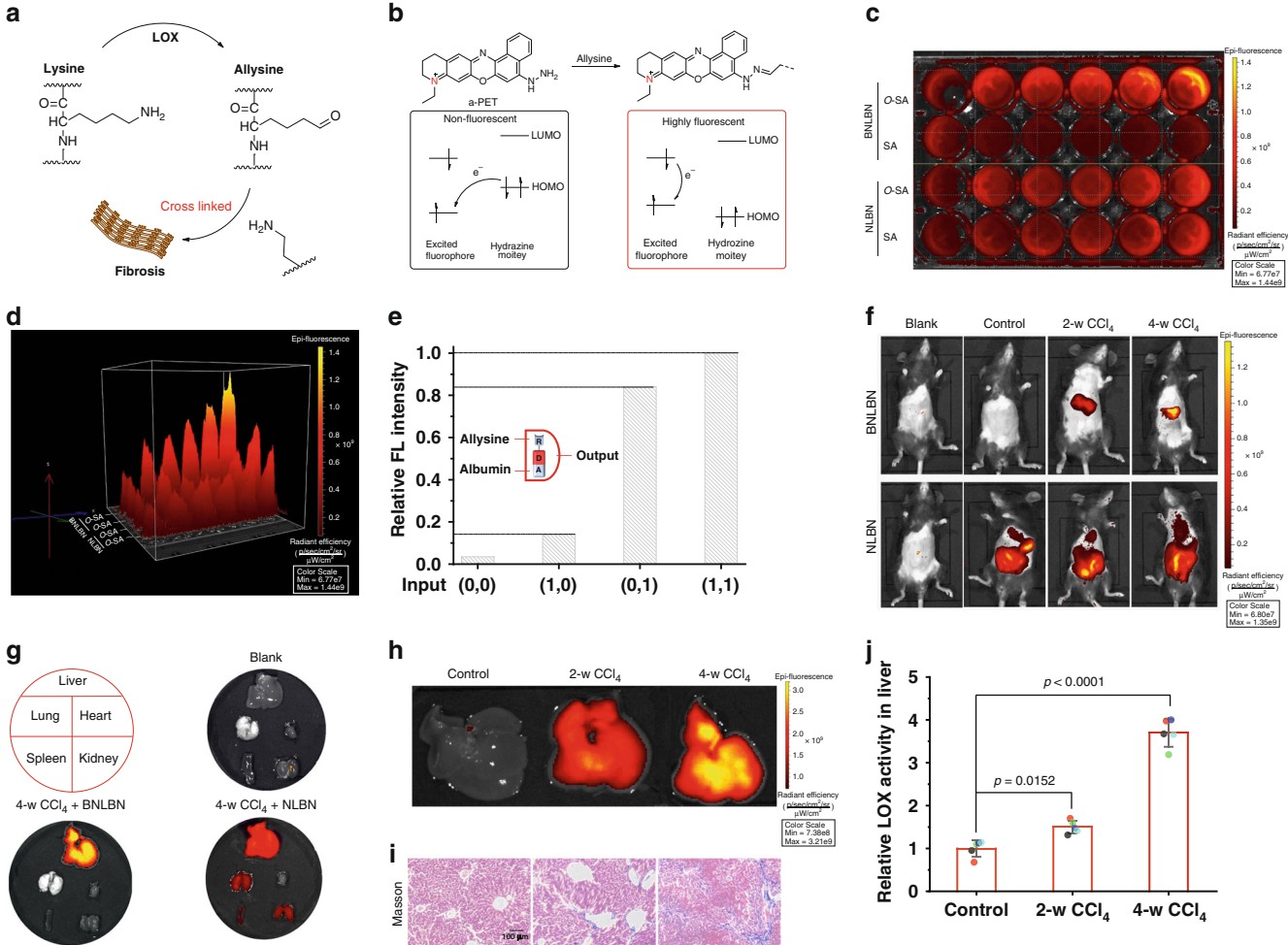

**Fig. 4 Evaluation of the devised "pocket-escaping" probe for in vivo imaging of liver fibrosis in mice. a** The illustration of fibrosis mediated by allysine. **b** The mechanism of developing probe for allysine. **c, d** Fluorescence photograph and quantification analysis of BNLBN and NLBN (10 μM for each) in imaging O-SA (0−10 μM). **e** Molecular logic gates for illustrating the backdoor problem of NLBN on the detection. **f** In vivo tracing liver fibrosis by BNLBN and NLBN (100 μL, 200 μM, intravenously for each of these two probes). **g** Tissue fluorescence images with BNLBN and NLBN in liver, lung, heart, spleen, and kidney. **h** Fluorescence imaging of liver in different model by BNLBN. **i** Masson staining of liver tissues from mice. Scale bar = 100 μm. **j** LOX activity assay of CCl₄-induced liver fibrosis model. Data are represented as mean values ± SD ($n = 5$ mice for each group). Statistical significance was determined by one-way ANOVA with Tukey's test. $p = 0.0152$ for 2-w CCl₄ treated group versus the control group. $p < 0.0001$ for 4-w CCl₄ treated group versus the control group. blank: without any treatment; control: treated with PBS; mild fibrosis: twice a week, intraperitoneal injection of CCl₄ (10% in olive oil; 1 mL/kg, for 2 weeks); severe fibrosis: the same treatment as but for 4 weeks. Source data for Fig. 4j are provided as a Source Data file.

liver-related cell types, stellate cell line HSC-T6 and hepatocytes cell line AML 12 (Supplementary Fig. 15). Then, we evaluated the in vivo efficacy of BNLBN for detecting fibrogenesis in a murine model of liver fibrosis induced by carbon tetrachloride (CCl₄)[40,41]. The mice were divided into four groups—(i) blank: without any treatment; (ii) control: treated with PBS; (iii) mild fibrosis: twice a week, intraperitoneal injection of CCl₄ (for 2 weeks); (iv) severe fibrosis: the same treatment as (iii) but for 4 weeks. Then, the animals were injected with BNLBN or NLBN and, after 4 h, anesthetized for subsequent imaging and histological analysis.

As shown in Fig. 4f, BNLBN successfully detected hepatic fibrogenesis; it generated intensive fluorescence in the sample groups (iii) and (iv), while producing no signal in the blank (i) and control (ii) groups. In contrast, NLBN released strong fluorescence in all groups (ii−iv) where it was presented. Further, analysis of the organ distribution of the fluorescence revealed that false signal from NLBN was even observed in the lung, spleen, and kidney tissue (Fig. 4g). The signal intensity of BNLBN correlated well with the extent of fibrosis, according to histological and biochemical assessments (Supplementary Fig. 19).

A stronger fluorescence signal was observed in group (iv) than group (iii). These data are consistent with morphological observation[42,43] and Masson's trichrome staining, which revealed that the formation of collagen fibers progressed 2 weeks and strengthened 4 weeks after CCl₄ injection, in contrast to the control group (Fig. 4g, i). Also, the liver LOX activity increased by 1.5 and 3.7 folds, after 2 and 4 weeks of CCl₄ treatment, respectively (Fig. 4j). Collectively, these findings suggest that BNLBN can effectively detect allysine and therefore image liver fibrosis in vivo with high accuracy.

Finally, to evaluate the translational potential of the designed fluorophore, we examined its whole-body distribution, pharmacokinetics, and toxicity in vivo. Measurement of their distributions in major organs at 1, 3, and 6 h after intravenous injection showed that BNLBN mainly accumulated in the liver and kidney, accounting for over 60% of the injected dose at 3 h post administration, while NLBN was in the liver, kidney, and intestine (Supplementary Fig. 20). Both BNLBN and NLBN were distributed rapidly in the bloodstream. As shown in Supplementary Fig. 21, the distribution half-lives ($t_{1/2\alpha}$) of BNLBN and

NLBN were similar. The terminal half-life ($t_{1/2\beta}$) of NLBN followed a different trend, about 2.7-fold longer than BNLBN, likely due to its "pocket-capturing" by serum albumin[44,45]. The blood circulation seemed closely related to the physicochemical properties of a probe, especially its binding affinity for the serum albumin. To further investigate the safety of BNLBN and NLBN, the mice were intravenously injected with probes (200 µM, 100 µL for each) every other day for 4 days and monitored for up to 14 days[46]. No weight loss (Supplementary Fig. 22a), pathological changes to the five major organs (the heart, liver, spleen, lung, and kidney, Supplementary Fig. 22b), or abnormal biochemical indexes—including serum levels of alanine aminotransferase, aspartate aminotransferase, blood urea nitrogen, and lactate dehydrogenase (Supplementary Fig. 22c)—were observed after the administration of BNLBN and NLBN, highlighting the in vivo safety of BNLBN and NLBN at the experimental doses.

## Discussion

In this study, we have addressed a critical backdoor problem for the in vivo application of a broad type of NIR dyes—that is, probes constructed on such dyes bind to albumin and trigger interfering fluorescence in imaging. Although this phenomenon is utilized elegantly for prolonging the circulation or improving the stability of nanocomplex-based imaging tools[8–14], it is essentially more a problem than a benefit in practice. This interaction may substantially affect a vast number of NIR-type molecular dyes designed under the TICT principle for detection of specific analytes; and the interfering signal is usually intense to mask the expected signal and lead to inaccurate detection[6,26]. However, the mechanism underlying the interference was unknown and hence no effective solution was available. In the present study, we revisited the fundamental of the backbone structure of NIR fluorophores. It is an acceptor−π−donor (A-π-D) structure, with the electron transfer from D to A as the basis for fluorescent emission. The twist of the D/A pair, together with a reasonably extended π-conjunction, determines emission wavelength[19]. Using both experimental and docking methods, we uncovered the key mechanism for this problem: the A group mediates the capture of itself—hence the dye—into the albumin pockets (pocket-capturing). Our results showed that blocking the A group to be a coplanar structure can avoid dye trapping into the albumin pockets that gave rise to a pocket-escaping NIR dye. Based on the pocket-escaping NIR fluorophore, a novel probe was developed that successfully prevented the interference of unspecific signal caused by albumin in the imaging of liver fibrosis in a mouse model. The in vivo findings, in agreement with the in vitro data, highlight the severity of albumin interference and validate the effectiveness of re-designing a pocket-escaping A group in preventing this backdoor problem.

In elucidating the above mechanism, we synthesized five NIR fluorophores and assessed their fluorescent activation upon interaction with albumin. These five fluorophores represent the three most common types of TICT-based dyes in practical use, namely dicyanomethylene-4H-pyran (DCM), Nile blue, and Cyanides dyes[28–30]. Interestingly, some of them were previously reported to bind albumin and exert fluorescence[26,35], but there was no systemic investigation into the commonness of the phenomenon and its in-depth mechanism is required. Our data indicated that albumin—of different origins of human, bovine, and mouse—and FBS enhanced the fluorescence of these commonly used, A−π−D NIR dyes, by targeting the twistable A group on the dye molecule. The strong π−π stacking by electronic coupling between A group and the Tryptophan (Trp) of albumin pocket formed an albumin−fluorophore complex (pocket-capturing). This interaction held the twisted conformation of NIR

dyes, enhancing the TICT process and resulting in fluorescence enhancement due to a decreased internal rotation of the dyes (the nonradiation process)[8]. This is potentially a mechanism for most probes in interaction with albumin[5,6,26,27]. Subsequently, we transformed a structure insensitive to albumin into sensitive ones by introducing different A groups, confirming this pocket-capturing principle to be generic.

Inspired by our above findings, we designed a type of A group that was blocked to be a coplanar structure. Combining the binding assay by tryptophan fluorescence measurements[35] with fluorescence titration upon the addition of different concentration of albumin, our data suggested that this modification could avoid fluorophore trapping into the albumin pockets that gave rise to a pocket-escaping NIR fluorophore. These data highlighted that blocking the twisted A group was an effective method to promote the escaping from the pocket's capture by albumin. Based on the pocket-escaping NIR fluorophore, a novel probe was developed that successfully prevented the interference of unspecific signal caused by albumin in the detection of liver fibrosis in a mouse model, while the control probe developed based on conventional strategy failed. That the probe displayed false fluorescence signal may have two reasons. First, it can be attributed to the backdoor problem based on its designing strategy. Second, this pocket-capturing may affect its chemical activity in the detection of allysine in vivo, for similar results were observed in protein corona, which can hinder the targeting capacity of nanoparticles[47]. Meanwhile, our results agree with previous findings to confirm allysine as a potential marker for fibrosis[36–39]. Further, NLBN had a higher content than BNLBN in the intestine, indicating that the NLBN−albumin complex mostly went through the hepatobiliary clearance route, which is commonly known for mediating the metabolism of plasma proteins, such as albumin[48]. To our knowledge, the pocket-escaping design NIR probe is the first of its kind that can image different stage of liver fibrosis and achieve enhanced accuracy of imaging in vivo in the abundant presence of albumin.

There are at least two major directions for future research. First, it will be interesting to design structures to control the albumin−dye binding, taking into consideration charge and hydrogen bonding interactions with the dye[33]. This can provide a structural framework for interpreting the binding of dyes to biomolecules. Second, based on the findings, we can develop a pocket-escaping NIR platform by designing more NIR probes with this advantage for in vivo applications and test them in more clinically relevant models. As such, our mechanistic explanation and effective solving of this generic backdoor problem may provide fresh insights for designing imaging tools for broad biomedical applications.

## Methods

**Photophysical properties**. For photophysical characterization, compounds were dissolved in dimethyl sulfoxide to prepare a stock solution (5 mM), which were further diluted to 10 µM as the testing concentration in PBS solution (pH 7.4, 10 mM).

**DFT calculations**. All the calculations were carried out using the Gaussian 09 program package. All the geometries of these dyes were optimized at B3LYP/6-31 + G(d) level. The MO energy levels were computed at the same level of theory.

**Animal experiments**. The animal protocols were reviewed and approved by the Animal Care and Use Committee of Nanjing University and University of Macau. All female C57BL/6 mice (~20 g) were operated in accordance with institutional ethics committee regulations and guidelines on animal welfare.

**CCl$_4$-induced liver fibrosis model**. Ten-week-old C57BL/6 female mice hepatic fibrosis was induced using carbon tetrachloride (CCl$_4$). CCl$_4$ was diluted to 10% in olive oil and introduced by intraperitoneal injections at dosages of 0.5 µL pure CCl$_4$/g (100 µL/20 g mouse) body weight twice a week, for 2 or 4 weeks. The

control group was treated with a same volume of PBS buffer (pH 7.4, 10 mM) and the group without any operation was as a blank control.

**Molecular docking study**. The three-dimensional structures of TICT dyes were constructed using Chem. 3D ultra 14.0 software (Chemical Structure Drawing Standard; Cambridge Soft Corporation, USA (2016)). The crystal structures of protein domain were downloaded from the RCSB Protein Data Bank (http://www.rcsb.org). Molecular docking of all compounds was performed via Discovery Studio (version 3.5) as implemented through the graphical user interface CDOCKER protocol.

**Statistics**. Data are presented as mean ± standard deviation (SD). Group sizes ($n$) are indicated in figure legends. Significance was assessed by the ordinary one-way analysis of variance (ANOVA) with Tukey's honestly significant difference (HSD) post hoc evaluation.

All other methods and protocols are provided in the Supplementary Information.

**Reporting summary**. Further information on research design is available in the Nature Research Reporting Summary linked to this article.

## Data availability

The data of this study are available from the corresponding author upon reasonable request. Protein Data Bank (http://www.rcsb.org) accession codes are 2BXC and 2BXM. The source data underlying Figs. 1d, e, 2e, f, 3e, f, 4j, Supplementary Figs. 6, 7, 14, 15, and 21 are provided as a Source Data file.

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

## Acknowledgements

We would like to thank Prof. Ru Yan and Dr. Yushun Yang for technical assistance in the preparation of the paper. This study was funded by the Science and Technology Development Fund, Macau SAR (File No. 080/2016/A2, 0018/2019/AFJ & additional fund to State Key Laboratory of Quality Research in Chinese Medicine) and the University of Macau Research Committee (MYRG2017-00028-ICMS, MYRG2019-00080-ICMS).

## Author contributions

C.W. and P.X. designed the study. P.X. performed major chemical experiments. Y.N., R.M., Z.W., D.X., and H.L. performed all biological experiments and some chemical experiments. P.X., L.D. and C.W. drafted the manuscript. C.W. provided funding supports. All authors contributed to data analysis and manuscript drafting.

## Competing interests

The authors declare no competing interests.
