## [Peer Review File · Nature Communications]

Reviewers' comments:

Reviewer #2 (Remarks to the Author):

In this work, the authors reported a strategy to avoid the encapsulation of the A group of the dye in the hydrophobic pocket of albumin, which enhanced the imaging accuracy for liver fibrosis in vivo. Overall, this work is interesting and well-written. However, there are some questions that need to be addressed.

1. The authors said that "The interference of albumin for the in vivo imaging of NIR probes can be a common issue but has to date been mostly overlooked. " Whether the binding between albumin and dye would interfere with the imaging results depends on the imaging target. Albumin binding may be beneficial by extending the circulation of dye to enhance the accumulation in certain imaging target, such as tumor.
2. The improved imaging efficiency was demonstrated by the comparison between BNLBN and NLBN. However, only the fluorescence photograph was given. Their emission spectrum was not shown. If their emission peak positions are different, it might generate different background signal due to the penetration or scattering differences of dyes.
3. NLBN generated stronger background signal in intestinal tract than BNLBN. Is there any difference regarding the clearance pathway of dyes after modification of chemical structure? The stronger signal in intestinal tract may be due to faster hepatobiliary clearance of a dye.
4. It seems that BNLBN is more responsive to O-SA than NLBN. Is it possible that this stronger responsiveness is responsible for generating less background signal in intestinal tract?
5. The time-dependent in vivo imaging data should be presented for comparison.
6. The binding energy of NLB and BNLB to albumin should also be provided.
7. The fluorescence photograph of BNLBN and NLBN lacks the concentration information.
8. The in vitro time-dependent responsiveness of BNLBN and NLBN should also be provided.

Reviewer #3 (Remarks to the Author):

The manuscript, "A pocket-escaping design to prevent the common interference with near-infrared fluorescent probes in vivo", presents a novel idea for improving in vivo imaging using an albumin "pocket-escaping" dye structure. This design is inspired by the previous imaging result which most of fluorophores easily bind albumin, generating unspecific fluorescence to mask the specific signal. NIR dyes without "unspecific background signal" have many advantages for imaging and monitoring in alive body. Unfortunately, to review this manuscript thoroughly, major changes need to be done in advance. The manuscript should therefore not be published in its current form.

Detailed comments follow:

1. The details of cyanine dye binding to albumin have been investigated for over a decade. The authors do not acknowledge this body of work, therefore it is difficult to compare the present results on binding and fluorescence with the existing literature. For example, for binding site of dye with albumin, several latest publications in terms of NIR-II tail emission of cyanine dye and bivalent Evans blue derivative have confirmed the hydrophobic pockets of albumin. How does the present work compare? Is the systematic study of dye-protein binding methods novel? How does quantum yield compare to previous similar reports?
2. The ¹H NMR titration of different equivalent albumin is not clear.
3. In Fig. 1, the calculation method of fluorescence enhancement is not clear.
4. Explanation of the phenomenon where the intensity in FBS is higher than in PBS, is missing. The author also need calculate the optimal binding ratio, as other binding site may contribute the fluorescence enhancement.
5. The author also need investigate the temperature dependent fluorescence enhancement.
6. In Figure 2 –TICT induced orbital plots (LUMO and HOMO) are missing.
7. Albumin-fluorophore binding can efficiently enhance TICT process, which has been confirmed in

latest publication with IR-783/albumin as model system. Author here did not include the possible charge transfer between dye and albumin.

8. Evidence for "pocket-escaping" design is insufficient, the authors should add direct evidences, such binding affinity between NLB/BNLB and albumin.

9. To translate the design fluorophore to animal using, whole body distribution, pharmacokinetics and in vivo toxicity are necessary.

POINT-TO-POINT RESPONSES TO REVIEWERS' COMMENTS

Manuscript No. NCOMMS-19-30705

We highly appreciate the two experts for their valuable suggestions. In the past two months, we have added new data and revised our manuscript.

We are pleased to share our responses below.

Reviewer #2:

In this work, the authors reported a strategy to avoid the encapsulation of the A group of the dye in the hydrophobic pocket of albumin, which enhanced the imaging accuracy for liver fibrosis *in vivo*. Overall, this work is interesting and well-written. However, there are some questions that need to be addressed:

Q1. The authors said that “The interference of albumin for the *in vivo* imaging of NIR probes can be a common issue but has to date been mostly overlooked.” Whether the binding between albumin and dye would interfere with the imaging results depends on the imaging target. Albumin binding may be beneficial by extending the circulation of dye to enhance the accumulation in certain imaging target, such as tumor.

A1. We thank the reviewer for this comment and would like to share our opinions below:

- 1) We agree with the reviewer that the albumin-dye binding may have different impacts, depending on individual applications. As the reviewer points out, such interaction might bring benefits in certain cases. For example, researchers intentionally fabricated imaging materials and albumin together into nanocomplexes before *in vivo* administration (e.g. *Small* 2015,3932; *Adv Mater* 2015, 903; *Adv Mater* 2015, 6820; *Theranostics* 2017, 3667). These studies are impressive, providing new approaches for extending the circulation and/or improving the biocompatibility of the imaging tools.
- 2) But meanwhile, the dye-albumin binding is still a fundamental problem for the *in vivo* application of a wide range of molecular NIR probes. First, the abovementioned excellent works did not aim to solve this problem. Instead, they were focusing on improving the features of imaging materials that already exert fluorescence – bringing in higher stability, compatibility, or accumulation, among others. These methods did not, and did not aim to, reduce the interference during an off-on/on-off switch for molecular probes designed for analyte-specific detection. So, their and our studies have different goals. Our goal is to minimize the interfering signal caused by this binding, that would be crucial for maintaining the accuracy of a molecular probe – especially when sensitivity is highly demanded. Second, in practice, unlike pre-fabricating a nano-complex as mentioned above, for molecular probes without former conjugation, it is difficult to control their binding with albumin real-time *in vivo* and predict the consequence of enhancement or interference. Overall, the key problem, that the dye will interact with albumin *in vivo* to produce unspecific and intense fluorescence to mask the specific signal from the analyte for fluorescent probes, is still unsolved.
- 3) Therefore, we highly acknowledge the potential benefits (or other possibilities rather than interference), yet we believe that the dye-albumin interaction is, overall, more a problem

than a benefit. And this problem is common for molecular NIR probes. Thus, we tried to develop a general approach to resolve it. Our present study provides insights into the mechanism of the binding and, correspondingly, offers a possible solution to this problem, as evidenced by positive results from both in *vitro* and *in vivo* experiments.

- 4) Thanks to the reviewer's suggestion, we have softened our tongue in both the Introduction and Discussion of the revised article.

Q2. The improved imaging efficiency was demonstrated by the comparison between BNLBN and NLBN. However, only the fluorescence photograph was given. Their emission spectrum was not shown. If their emission peak positions are different, it might generate different background signal due to the penetration or scattering differences of dyes.

A2. This is an important reminder. We have assessed the capability of **BNLBN** and **NLBN** in detecting *O*-SA in PBS (10 mM, pH 7.4). As shown below, **BNLBN** and **NLBN** displayed similar emission peak at around 650 nm (665 nm for **BNLBN** and 655 nm for **NLBN**).

We have added this result as a new Supplementary **Figure S17**.

Figure R1 (also the new **Figure S17**). Fluorescent titration of **BNLBN** and **NLBN**. (A) Fluorescent spectrum of **BNLBN** (10 μM) with the addition of *O*-SA (0-10 μM) in PBS (10 mM, pH 7.4) and (B) the linear relationship between them; (C) Fluorescent spectrum of **NLBN** (10 μM) with the addition of *O*-SA (0-10 μM) in PBS (10 mM, pH 7.4) and (D) the linear relationship between them. Each spectrum was recorded after 5 min.

Q3. NLBN generated stronger background signal in intestinal tract than BNLBN. Is there any difference regarding the clearance pathway of dyes after modification of chemical structure? The stronger signal in intestinal tract may be due to faster hepatobiliary clearance of a dye.

A3. The reviewer is correct! To compare the difference of clearance of **BNLBN** and **NLBN**, we quantified the biodistribution, by intravenously injecting equivalent doses of probes and measuring the dye in different organs at 1, 3, and 6 h post-injection. As shown in the new **Figure S20**, we found that the probe content obviously decreased in the liver over time and gradually increased in kidney, indicating that **BNLBN** could be cleared from the liver and kidney. **NLBN** showed a similar tendency in these two organs with **BNLBN**. Interestingly, **NLBN** had a higher content in the intestine than **BNLBN**, which might suggest that the probe-albumin complex mostly went into the hepatobiliary clearance route, in agreement with literature (*Adv Healthcare Mater* 2016, 2510; *Nat Biotechnol* 2007, 1165).

Q4. It seems that **BNLBN** is more responsive to O-SA than **NLBN**. Is it possible that this stronger responsiveness is responsible for generating less background signal in intestinal tract?

A4. It is an excellent assumption and might be one of the reasons; yet interestingly, we found that a more apparent reason was in their different metabolism *in vivo* – as we answer to Q3 above and mention in the Discussion part. The ‘pocket-capturing’ **NLBN** went through the hepatobiliary clearance route that was faster than the ‘pocket-escaping’ **BNLBN** metabolized through hepatic and renal elimination. As we demonstrated, **NLBN** could bind to albumin (which might account for this type of clearance) and trigger fluorescence, which produced a higher background than **BNLBN** in the intestine. But certainly, the subtle responsiveness of two probes for O-SA and albumin should also play a role in the intestine and is worth a comprehensive future study.

Q5. The time-dependent *in vivo* imaging data should be presented for comparison.

A5. We have now added the data as a new Supplementary **Figure S18**. For **BNLBN**, significant fluorescence enhancement was observed in the liver area from 45 to 180 min after. For **NLBN**, the signal spread across a large area of the body throughout this period of observation.

Figure R2 (also the new **Figure S18**). Time-dependent imaging *in vivo*. **BNLBN** (A) and **NLBN** (B; 200 μ M in a 10 mM PBS solution, 100 μ L for each) were injected via tail vein. Images were captured at 45, 90, and 180 min.

Q6. The binding energy of **NLB** and **BNLB** to albumin should also be provided.

A6. Isothermal titration calorimetry (ITC) was performed to calculate the energetics of **NLB** interaction with albumin at 25°C. Our data showed that **NLB** binding to albumin was an endothermic

reaction (new **Figure S13**). The binding energy was dominated by a large positive entropy change ($\Delta H = 5.49$ kcal/mol), change in entropy ($T\Delta S = 13.3$ kcal/mol) and free energy change ($\Delta G = -7.8$ kcal/mol). The interaction of **BNLB** with albumin was undetectable.

Q7. The fluorescence photograph of BNLBN and NLBN lacks the concentration information.

A7. We are sorry for this missing. We have added it in the revised manuscript (**Figure 4D**).

Q8. The in vitro time-dependent responsiveness of BNLBN and NLBN should also be provided.

A8. We have added the time-dependent spectrum as a new Supplementary **Figure S16**.

Figure R3 (also the new **Figure S16**). Time-course response of **BNLBN** (A) and **NLBN** (B) to the addition of *O*-SA and SA. Reaction conditions: 10 μ M probe was mixed with 5 μ M *O*-SA/SA in a PBS (10 mM, pH 7.4) solution.

Reviewer #3:

The manuscript, “A pocket-escaping design to prevent the common interference with near-infrared fluorescent probes in vivo”, presents a novel idea for improving in vivo imaging using an albumin “pocket-escaping” dye structure. This design is inspired by the previous imaging result which most of fluorophores easily bind albumin, generating unspecific fluorescence to mask the specific signal. NIR dyes without “unspecific background signal” have many advantages for imaging and monitoring in alive body. Unfortunately, to review this manuscript thoroughly, major changes need to be done in advance. The manuscript should therefore not be published in its current form. Detailed comments follow:

Q1. The details of cyanine dye binding to albumin have been investigated for over a decade. The authors do not acknowledge this body of work, therefore it is difficult to compare the present results on binding and fluorescence with the existing literature. For example, for binding site of dye with albumin, several latest publications in terms of NIR-II tail emission of cyanine dye and bivalent Evans blue derivative have confirmed the hydrophobic pockets of albumin. How does the present work compare? Is the systematic study of dye-protein binding methods novel? How does quantum yield compare to previous similar reports?

A1. We thank the reviewer for the insightful questions.

- 1) We are pleased to share the same findings on the binding between albumin and NIR dyes. Both our work and previous studies provide useful insights to this area. The reviewer is correct that albumin-dye binding has been studied for many years. Recently, the albumin-dye complex has been utilized to increase the circulation time, quantum yield, stability, and imaging contrast, as imaging agent in *in vivo* albumin labelling or other tissue imaging (e.g. *Biochemistry* 2011, 2691; *Adv Mater* 2018, 1802546; *Nat Commun* 2017, 15269). We have cited these studies in the revised manuscript.
- 2) We appreciate the previous work on the binding of cyanine dye and Evans blue derivative with the hydrophobic pockets of albumin (*Sci Adv* 2019, eaaw0672; *P Natl Acad Sci* 2015, 208; *J Nucl Med* 2014, 1150). Our present work adds weight to the previous studies, not only extending the explanation to a range of TICT NIR dyes and revealing the commonness of this mechanism with both experimental and docking approaches, but more importantly proposing a new, practical, engineered strategy to fundamentally preventing the binding – that is, changing the ‘A’ group for ‘pocket-escaping’. We have thankfully cited these three papers in the revised paper.
- 3) Yes, our systematic investigation of the dye-protein binding methods has clear novelty. Inspired by the previous works, our present study focuses on the molecular structure of NIR dyes, using both experimental and docking methods to identify an A group-mediated capture of the dyes into the albumin pockets (‘pocket-capturing’) as the key mechanism for the binding between albumin and dyes. As mentioned above, we build on this mechanistic finding to engineer a new and generic approach for increasing the accuracy of NIR imaging probes.
- 4) At last, we measured the quantum yield of the five NIR dyes in 100 μ M BSA solution, using cresol purple as reference ($\Phi_f = 0.58$ in ethanol). These dyes show comparable quantum yield (DCM, 4.3%; NLB, 2.1%; Cy7, 1.8%; HCy7, 2.7%; Cy5, 19.2%) with reported ones (*Nat Commun* 2017, 15269; *Angew Chem Int Edit* 2018, 7483; *Adv Healthc Mater* 2018, 1800589).

Q2. The ¹H NMR titration of different equivalent albumin is not clear.

A2. We have improved the figure quality and added part of the data in the revised manuscript. Please refer to the new Supplementary **Figure S8** and the revised **Figure 1F**.

Q3. In Fig. 1, the calculation method of fluorescence enhancement is not clear.

A3. We defined fluorescence enhancement as the ratio of the fluorescence intensity of a dye in each condition to that in PBS. We have added this into the revised manuscript.

Q4. Explanation of the phenomenon where the intensity in FBS is higher than in PBS, is missing. The authors also need calculate the optimal binding ratio, as other binding site may contribute the fluorescence enhancement.

A4. We appreciate this insightful advice and are delighted to share our opinions below:

- 1) We measured the intensity of **DCM**, **NLB**, **Cy5**, **HCy7**, and **Cy7** in FBS. For the same dye, it has a stronger fluorescence enhancement in FBS than in PBS, comparable to the intensity in 300-600 μ M BSA. This result agrees with the findings in literature, where the albumin

concentration in FBS ranged from 19 to 32 mg/mL (equivalent to 286-481 μM) (*Toxicology*, 2002, 201). We have added the data as a new Supplementary **Figure S6**.

Figure R4 (also new **Figure S6**). Fluorescence enhancement of **DCM**, **NLB**, **Cy5**, **HCy7**, and **Cy7** (10 μM for each) in different concentration of BSA (300, 600 μM) and FBS. The data were determined by the ratio of the fluorescence intensity of dyes in different conditions to that in 100 μM BSA.

- 2) Back to our manuscript, we first validated the albumin-mediated fluorescence enhancement in TICT NIR fluorophores, and then suggested that the mechanism was albumin binding the dyes by “capturing” the A groups into its hydrophobic pockets. Here, the strong π - π stacking by electronic coupling between A groups of fluorophores and the Tryptophan (Trp) of albumin pockets created the albumin-fluorophore complex (pocket-capturing). This interaction held the twisted conformation of NIR fluorophores, causing fluorescence enhancement due to a decreased internal rotation of the dyes (the nonradiation process; *Sci Adv* 2019, eaaw0672).
- 3) Fluorescent spectrum of **DCM**, **NLB**, **Cy5**, **HCy7**, and **Cy7** was tested in protein solutions ranged from 1:1 to 60:1 protein-to-dye molar ratio. Please see figure below, where our data showed that it was a concentration-dependent enhancement, with no evident saturation found in this range of ratio. Combining the temperature effect data (new **Figure S7**) and ITC data (new **Figure S13**), we confirmed that this binding is entropy-driven and increasing temperature enhanced the binding. Our opinion is that this protein-dye binding at room temperature is not as effective as at higher temperature, and higher concentrations of the protein also contribute to this binding. In this part, we attempted to simulate physiological environment concentration of albumin and explore the binding mechanism of these dyes. Thus, we believe there is no optimal binding ratio in this scenario.

Figure-for-response-letter-only 1 Fluorescence enhancement of **DCM**, **NLB**, **Cy5**, **HCy7**, and **Cy7** ($10\ \mu\text{M}$ for each) with different ratio of HSA (1:1, 1:5, 1:10, 1:30, and 1:60). The data was determined by the ratio between the fluorescence intensity of dyes in protein and that in PBS.

- 4) Finally, we agree with the reviewer that different binding sites may contribute to the fluorescence enhancement. As shown in **Figure S10-12**, a competitive assay indicated the binding sites on albumin for these “transplanted” TICT dyes to be: i) IB, IIA, and IIB for **NC**; ii) IIA and IIB, for **BNCN**; iii) IIA and IIB for **BNCY**; iv) IB, IIA, and IIB for **BNCE**.

Q5. The authors also need investigate the temperature dependent fluorescence enhancement.

A5. We have tested the temperature dependence of the NIR fluorophores by heating each dye from room temperature to $80\ ^\circ\text{C}$ in the presence of BSA. As shown below, as the temperature went up, the fluorescent enhancement increased and reached a plateau at around $70\ ^\circ\text{C}$, in agreement with literature (*Nat Comm* 2017, 15269 and *Sci Adv* 2019, eaaw0672) and the ITC result. We have added the data as a new Supplementary **Figure S7**.

Figure R5 (also new **Figure S7**). Fluorescence enhancement of **DCM**, **NLB**, **Cy5**, **HCy7**, and **Cy7** ($10\ \mu\text{M}$ for each) mixed with $100\ \mu\text{M}$ BSA at different temperatures. Each spectrum was recorded after heating for 10 min.

Q6. In Figure 2 –TICT induced orbital plots (LUMO and HOMO) are missing.

A6. The rotation of the malononitrile moiety on **BNCN** (with a dihedral angle of 90°) created an asymmetric π -conjugated backbone (**BNCN-90°**) that resulted in charge redistribution and further induced the TICT. The DFT data showed that TICT effectively lowered the energy gap between HOMO-LUMO from 3.05 eV on **BNCN** to 0.94 eV on **BNCN-90°**, matching well with the fluorescent result. We have added the data in the revised **Figure 2D**.

Figure R6 (also new **Figure 2D**). DFT optimized molecular orbital plots (LUMO and HOMO) of **BNC**, **BNCQ**, **BNCN**, and **BNCN-90°**.

Q7. Albumin-fluorophore binding can efficiently enhance TICT process, which has been confirmed in latest publication with IR-783/albumin as model system. Author here did not include the possible charge transfer between dye and albumin.

A7. We appreciate the reviewer for sharing the potential mechanism explanation, which we totally agree on. The strong π - π stacking by electronic coupling between A groups of fluorophores and the Tryptophan (Trp) of pocket on albumin results in albumin-fluorophore complex (pocket-capturing). As reported (*Sci Adv* 2019, eaaw0672), HOMO-1 of TICT dye lies in the indole ring of Trp. The electrons transfer from HOMO-1 to LUMO upon vertical excitation, fall to HOMO exerting emission, and return to HOMO – this process enhances TICT. We believe this theory is applicable in our model. In **Figure 3E**, our data also confirmed the interaction between Trp and fluorophore. The binding assay which was determined by Trp's fluorescence intensity indicated that **BNLB** (pocket-escaping dye) had a much lower binding affinity for albumin ($K = 4,400 \text{ M}^{-1}$) compared with **NLB** (pocket-capturing dye, $K = 38,910 \text{ M}^{-1}$). We have added relevant discussion in the 2nd paragraph of the Discussion section in the revised manuscript.

Q8. Evidence for “pocket-escaping” design is insufficient, the authors should add direct evidences, such binding affinity between NLB/BNLB and albumin.

A8. We thank the reviewer for this great reminder. We have performed two assays to illustrate the binding:

- 1) We first employed isothermal titration calorimetry (ITC) for quantitatively analysing the albumin-dye binding. As shown below, albumin (500 μM) was titrated into a 100 μM

solution of **NLB**. The binding isotherm revealed an endothermic binding characteristic with a calculated K_d of 161 μM . By contrast, **BNLB** did not bind to albumin under these conditions. We have added the data as a new Supplementary **Figure S13**.

Figure R7 (also new **Figure S13**). Quantitative measurements of **NLB** and **BNLB** binding to albumin. (A) Quantification of **NLB** binding to albumin by ITC. Albumin (500 μM in the syringe) was injected into a 100 μM solution of **NLB**. The data were fitted with a simple one-site-binding model, yielding a K_d of 161 μM . (B) Quantification of **BNLB** binding to albumin by ITC. Albumin (500 μM in the syringe) was injected into a 100 μM solution of **BNLB**. No binding was observed.

- 2) We next used a native Polyacrylamide Gel Electrophoresis (native PAGE) to characterize the binding. As shown below, the fluorescent signal of the **NLB**-albumin complex was observed, while that of the **BNLB**-albumin complex was not detected. We have added the data as a new Supplementary **Figure S14**.

Figure R8 (also new **Figure S14**). Electrophoresis analysis of **NLB** and **BNLB** binding to albumin on 8% native PAGE. The gels were imaged under fluorescence (red band) and then stained with Coomassie brilliant blue to visualize proteins (blue band). Comparison of the addition of SDS (A) or not (B) in the electrophoresis of free dyes.

- 3) The above data provide direct evidence from two aspects for validating the “pocket-capturing” and “pocket-escaping” property of **BNLB** and **NLB**, respectively.

Q9. To translate the design fluorophore to animal using, whole body distribution, pharmacokinetics and in vivo toxicity are necessary.

A9. These are excellent suggestions! We have added new data in three aspects:

- 1) Biodistribution: equivalent doses of **BNLBN** and **NLBN** were intravenously injected and their contents in major organs were measured at 1, 3, and 6 h. **BNLBN** was mainly in the liver and kidney, accounting for over 60% of the injected dose at 3 h after administration. **NLBN** was easily trapped in the liver, kidney, and intestine, indicating that “pocket-escaping” design also changed the *in vivo* biodistribution of probes. We have added the data as a new Supplementary **Figure S20**.

Figure R9 (also new **Figure S20**). Quantitative analysis of **BNLBN** and **NLBN** in major organs (heart, liver, spleen, lung, and kidney) at 1, 3 and 6 h post-injection.

- 2) Pharmacokinetics: both **BNLBN** and **NLBN** were found to be distributed rapidly in the bloodstream, with no significant difference in the distribution half-life ($t_{1/2\alpha}$). The terminal half-life ($t_{1/2\beta}$) of **NLBN** was about 2.7-fold longer than **BNLBN**, likely due to the “pocket-capturing” by serum albumin (e.g. *Adv Healthcare Mater* 2016, 2510; *Nat Biotechnol* 2007, 1165). The blood circulation is closely related to the physicochemical properties of each probe, especially the binding ability with serum albumin. We have added the data as a new Supplementary **Figure S21**.

Figure R10 (also new **Figure S21**). Pharmacokinetics of **BNLBN** and **NLBN** in mice. (A) Time-concentration curves from 3 to 240 min; (B) Pharmacokinetic parameters in the blood. AUC, area under drug concentration-time curve.

- 3) Toxicity *in vivo*: the mice were intravenously injected with probes (200 μM , 100 μL for each) every other day for 4 days, and then monitored for up to 14 days (*Biomaterials* 2017, 130). No significant loss of body weight was found. No lesions (necrosis, edema, inflammation, or

hyperplasia) were observed in the sections of the five organs (heart, liver, spleen, lung, and kidney). No abnormality was recorded in the blood serum analysis for alanine aminotransferase (ALT), aspartate aminotransferase (AST), blood urea nitrogen (BUN), and lactate dehydrogenase (LDH). Thus, the data suggest that **BNLBN** and **NLBN** had no toxicity in mice at the experimental dose. We have added the data as a new Supplementary **Figure S22**.

Figure R11 (also new **Figure S22**). *In vivo* toxicity study of **BNLBN** and **NLBN**. (A) Mice were weighed every three days after the injection of probes (200 μ M, 100 μ L for each); (B) Clinical chemistry indexes analysis of BUN, LDH, AST, and ALT; (C) H&E stained tissue slices (heart, liver, spleen, lung, and kidney). Scale bar: 100 μ m.

Finally, we appreciate both Reviewer 2 and Reviewer 3 for all the valuable suggestions, which have greatly helped us improve our study!

REVIEWERS' COMMENTS:

Reviewer #2 (Remarks to the Author):

I think this is an interesting design overall and from in vitro experiment, we can see how effective this strategy is to achieve more sensitive responsiveness to the target. Thus, I support the publication of this work with minor reversion.

It is hard to conclude the exact mechanism for improved performance of the fluorophore in vivo. As the authors discussed in the final part, other factors, such as targeting and clearance, may also be responsible for the improved performance. It would be better if the author can include the related information in the introduction to explain the advantages for albumin escaping strategy. I suggest the authors to include the "twisted intramolecular charge transfer" and "turn on responsiveness" into the first paragraph of introduction. It seems that this strategy is more effective for this kind of fluorophore. It may not be a general problem for other types of fluorophore. The authors said that "However, recent findings reveal that NIR fluorophores/probes can, unexpectedly, bind to serum albumin, and this binding generates fluorescence"

Reviewer #3 (Remarks to the Author):

The authors addressed the issues and now I agree with publication

POINT-TO-POINT RESPONSES TO REVIEWERS' COMMENTS

Manuscript No. NCOMMS-19-30705A

Reviewer #2 (Remarks to the Author):

I think this is an interesting design overall and from in vitro experiment, we can see how effective this strategy is to achieve more sensitive responsiveness to the target. Thus, I support the publication of this work with minor reversion.

Q1. It is hard to conclude the exact mechanism for improved performance of the fluorophore in vivo. As the authors discussed in the final part, other factors, such as targeting and clearance, may also be responsible for the improved performance. It would be better if the author can include the related information in the introduction to explain the advantages for albumin escaping strategy.

A1. We thank the reviewer for this great suggestion. We have added the related information in the Introduction of the revised article.

Q2. I suggest the authors to include the “twisted intramolecular charge transfer” and “turn on responsiveness” into the first paragraph of introduction. It seems that this strategy is more effective for this kind of fluorophore. It may not be a general problem for other types of fluorophore. The authors said that "However, recent findings reveal that NIR fluorophores/probes can, unexpectedly, bind to serum albumin, and this binding generates fluorescence"

A2. Good suggestion! We have modified the expression in the Introduction of the revised article.

Reviewer #3 (Remarks to the Author):

The authors addressed the issues and now I agree with publication.

We thank the reviewer for his/her insightful advice.

Finally, we thank again for all the valuable suggestions from the reviewers and editor throughout.